



# The E3SM version 1 Single Column Model

Peter A. Bogenschutz[1], Shuaiqi Tang[1], Peter Caldwell[1], Shaocheng Xie[1], Wuyin Lin[2], and
Yao-sheng Chen[3,4,5]

[1]Lawrence Livermore National Laboratory, Livermore, CA
[2]Brookhaven National Laboratory, Upton, NY
[3]Cooperative Institute for Research in Environmental Sciences, Boulder, CO
[4]University of Colorado, Boudler, CO
[5]Chemical Sciences Division, NOAA Earth System Research Laboratory, Boulder, CO

**Correspondence:** Peter A. Bogenschutz (bogenschutz1@llnl.gov)

**Abstract.** The single column model (SCM) functionality of the Energy Exascale Earth System Model version 1 (E3SMv1) is described in this paper. The E3SM SCM was adopted from the SCM used in the Community Atmosphere Model (CAM), but has evolved significantly since then. We describe changes made to the aerosol specification in the SCM, idealizations, and developments made so that the SCM uses the same dynamical core as the full general circulation model (GCM) component.

5    Based on these changes, we describe and demonstrate the seamless capability to "replay" a GCM column using the SCM. We give an overview of the E3SM case library and briefly describe which cases may serve as useful proxies for replicating and investigate some long standing biases in the full GCM runs, while demonstrating that the E3SM SCM is an efficient tool for both model development and evaluation.

## 1 Introduction

Despite advances in computation allowing for General Circulation Models (GCMs) to be run with progressively finer resolution with each successive generation, the parameterized physics in the atmospheric components of these GCMs have steadily become more complex. Indeed, while this increase in complexity often leads to better climate simulations due to more realistic and comprehensive processes being accounted for, understanding interactions between these parameterizations and the GCM

15   dynamics can be a daunting task. A tool to help GCM physics development and evaluation is the so-called Single Column Model (SCM) framework, which is a functionality that exists in many state-of-the-art GCMs. This work was pioneered by Betts and Miller (1986), with the link between SCMs, observations, and GCMs studied more extensively in Randall et al. (1996). A SCM is a mode where a single column of the atmosphere is run in isolation with prescribed atmospheric dynamics. Thus, the SCM will simulate unresolved processes within the atmospheric column, such as clouds, microphysics, turbulence,

20   and radiation while removing the complexity of the dynamics-physics interactions.





SCMs are often the first step in the GCM parameterization development and/or implementation process. This is due to the fact that SCMs can provide a framework for quicker and easier debugging compared to the full GCM counterpart. In addition, depending on the regime of interest being targeted, the SCM simulation can readily be compared against observations or large eddy simulation (LES). This allows for rapid feedback of the parameterization performance in a more process oriented environment. Park et al. (2014) and Bogenschutz et al. (2012) are examples of how an SCM is used to implement and evaluate new and complex families of parameterizations in the National Center for Atmospheric Research's (NCAR's) Community Atmosphere Model version 5 (CAM5; Neale et al. 2012) SCM. The SCM can also be used as a tool to explore configurations which may not be feasible to do in a full GCM. For example, Bogenschutz et al. (2012) explored CAM, with two different physics packages, with very high LES-like vertical resolution that would have been computationally burdensome to do with a full GCM run.

SCMs are also useful tools for examining GCM physical parameterization performance at the process level. For instance, Zheng et al. (2017) used CAM's SCM to diagnose the cause of a cloudy planetary boundary layer oscillation, which was found to be the result of coupling issues between the turbulence and microphysics schemes. Zhang and Bretherton (2008) performed SCM studies to show that cloud feedbacks in CAM3 were controlled by unphysical oscillations caused by interactions between convection and resolved-scale processes. The SCM also provides a useful tool to perform perturbed parameter sensitivity studies for complex parameterizations that contain an abundance of tunable parameters, such as those performed by Guo et al. (2015).

The Energy Exascale Earth System Model version 1 (E3SMv1; Golaz et al. 2019) is an Earth system model designed with funding by the Department of Energy (DOE) for research and applications relevant to its mission. While E3SMv1 contains three new components (ocean, sea ice, and river) that have not previously been coupled to an Earth system model, the atmosphere and land components were branched from the Community Earth System Model version 1 (CESM1; Hurrell et al. 2013), but have evolved since (Xie et al. 2018; Rasch et al. 2019). Therefore, E3SMv1 inherited CAM and its associated SCM (Gettelman et al. 2019), but its SCM has also evolved. Those changes will be documented in this paper.

While SCMs have demonstrated that they are valuable tools for parameterization development, testing, and process-oriented evaluation efforts, they are unable to elucidate remote impacts or clarify physical-dynamical interactions where three-dimensional transport effects come into play. In addition, the SCM may replicate the behavior of the full GCM better for certain regimes and conditions than others, though this issue is poorly understood and not well studied. In this paper we will preliminarily demonstrate the conditions under which the E3SM-SCM can serve as a useful proxy for GCM performance.

This paper will be organized as follows: In section 2 we describe the E3SM SCM and discuss modifications made to it since branching from CAM's SCM. The SCM case library is presented in section 3, with links to documentation provided for users to assist in running the model. Section 4 describes the "Replay" option, which allows the user to replicate a GCM column using the SCM for any point on the globe. Applications and examples of the SCM will be presented in section 5, including a preliminary analysis on when the SCM may serve as a useful proxy for the GCM and when it does not. Finally, a brief summary and discussion will be presented in section 6.





## 2 E3SM SCM Description

The E3SM Atmosphere Model version 1 (EAMv1; Xie et al. 2018; Rasch et al. 2019) was originally branched off from the National Center for Atmospheric Research's (NCAR's) Community Atmosphere Model (CAM). Therefore E3SM inherited the CAM SCM (Gettelman et al. 2019). However, several modifications were made to the E3SM SCM, which we document here.

### 2.1 Aerosol Specification

The E3SM model uses a prognostic modal aerosol model (Liu et al. 2016). While this represents a sophisticated and modern treatment of aerosol in E3SM, it presents challenges for a SCM, which have been noted in Lebassi-Habtezion and Caldwell (2015) (hereafter LHC2015). Chief among these is the fact that E3SM initializes all aerosol mass mixing ratios to zero, which results in unrealistically low aerosol concentrations until surface emissions loft sufficient aerosol. This is a process that can take several days to spin up (Schubert et al. 1979) and can significantly impact the simulated results of several SCM cases, which are only hours in duration. LHC2015 show that CAM5 simulations are very sensitive to the initialization of aerosol for stratiform boundary layer cloud cases, but not for shallow and deep convective cases (because deep and shallow convective microphysics schemes were not tied directly to the aerosol scheme). However, E3SM uses a unified treatment of shallow convection and planetary boundary layer turbulence (Golaz et al. 2002, Bogenschutz et al. 2013), thus the shallow convective clouds are tied to the large-scale microphysics scheme, which could lead to more severe impacts and sensivities for the shallow and deep convective cloud regimes if aerosol is not specified adequately.

We have implemented the three options proposed by LHC2015 to initialize aerosol in the E3SM SCM. The first option is to use prescribed aerosol climatology derived from a ten year E3SM present day simulation with climatologically prescribed sea surface temperatures (SSTs). The second option is to specify the droplet and ice concentrations in the microphysics, thus bypassing the aerosol-cloud interaction, and the third option is to use observed aerosol information from the intensive observation period (IOP) forcing file, if it is available. Selecting an aerosol specification option is mandatory for the E3SM SCM and a runtime error will result if a user attempts to run the SCM with no aerosol specification. For the scripts provided in the E3SM SCM case library (see section 3), the most appropriate aerosol specification is already set for each particular case. Should an E3SM user generate their own forcing and is unsure which option to select, we advise to use the prescribed aerosol specification as a default.

### 2.2 Idealizations

Many published LES comparison studies involving the simulation of boundary layer clouds include "idealizations". As an example, the goal of the LES intercomparison study of the Barbados Ocean and Meteolorological experiment (BOMEX; Holland and Rasmusson 1973) was to investigate the role of turbulence dynamics for the shallow cumulus boundary layer (Siebesma et al. 2002), while avoiding the complications of microphysics and radiation. As such, none of the LESs participating in the study included a microphysical parameterization in their simulation. In addition, the radiative heating tendencies for the





LES comparison were included in the large-scale forcing. Should an E3SM SCM user wish to evaluate the turbulence and cloud structure of the BOMEX case against the LES intercomparison study of Siebesma et al. 2002, not only would an apples-to-apples comparison would not be possible with an out-of-the-box configuration of the inherited SCM, but it would also be

scientifically invalid due to the fact that radiative tendencies would be double counted. While implementing these idealization switches into the model code is rather trivial, it is not an obvious task for the typical SCM user who is not familiar with the code and who may not be aware of the idealizations needed to match LES results.

   Therefore, with the goal of preventing improper case setups, we have implemented idealization switches into the E3SM SCM code to allow for apples-to-apples comparison with IOP forcings corresponding to the appropriate reference for the par-

ticular case (see section 3). The idealization switches added to the E3SM SCM framework includes idealizations related to turning off microphysics and radiation calculations. All relevant switches have been added by default to the run scripts for each particular case, but can be easily switched off by the user if they wish to examine that case using all E3SM physical parameterization schemes. Cases in the E3SM library which include idealizations turned on by default include: ATEX, BOMEX, DYCOMSRF01, DYCOMSRF02, MPACE-B, ARM shallow cumulus, and RICO (see table 1). The remaining cases have no

idealizations.

## 2.3  Consistent Dynamical Core

The code required to run the CAM SCM has long been entangled with the Eulerian dynamical core. As a result, the SCM couldn't be run with CAM's current Finite Volume (FV; Lin and Rood 1997) dynamical core or E3SM's Spectral Element (SE; Dennis et al. 2012) dynamical core. This is a problem because while the horizontal advection fields are provided by

the IOP forcing files, the dynamical core in the SCM still needs to compute the large scale vertical transport. The Eulerian dynamical core uses a simple Eulerian calculation for the large scale vertical transport, while the SE dynamical core uses a semi-Lagrangian method. Therefore, the inherited SCM was inconsistent with the GCM. In addition, there are stark differences in the numerics between the two dynamical cores; whereas the Eulerian core uses a leapfrog numerical scheme, the SE dynamical core uses a forward in time integration. This results in different coupling between the prescribed and computed dynamical forcing

with the physics and results in different dynamics and physics timesteps between the SCM and the GCM run, furthering the inconsistencies between the two configurations.

   Ideally, we want the SCM to be as close a proxy to the full GCM run as possible. Thus we upgraded the dynamical core to be the SE core for the E3SM SCM. The major challenge in achieving this goal was the fact that while it was possible to initialize the Eulerian dynamical core with one column, it is not possible to do so with the SE dynamical core which is made up

of a series of "elements" on a quadrilateral grid that forms the sphere. Within these elements lie the Gauss-Lobatto-Legendre (GLL) quadrature points, which also correspond to the location of the physics columns. The least invasive way to allow the SCM to work with the SE dycore was to "trick" the model by initializing the dynamical core at a low resolution configuration (we initialize the SCM at ne4 resolution, corresponding to a horizontal grid spacing of approximately 7.5° at the equator) but initialize only one physics column. Therefore, computation for physics parameterizations are only considered for one column





and we restrict the computation of the large-scale vertical advection (a one dimensional calculation) to only be done in the column of interest; thus the computational cost of the SE SCM is no different than that of the inherited SCM from CAM.

The upgrade to the SE dynamical core is also advantageous in the fact that E3SM no longer needs to maintain and support the Eulerian dynamical core, which was not used in any other model configuration. The biggest advantage, however, is the ability to seamlessly "Replay" a column of the full GCM with the SCM (see section 4).

## 3    SCM case Library


The E3SM SCM library is currently comprised of 25 cases, which range from widely used cases of idealized boundary layer cloud regimes of a few hours in duration to unique cases as that span the duration of years to a decade (i.e. continuous forcing from Atmospheric Radiation Measurement (ARM) Southern Great Plain (SGP) site; Xie et al. 2004). The list of available forcing files and their references can be found on tables 1 and 2. Cases such as DYCOMS, BOMEX, MPACE, RICO, and

ATEX are boundary layer cloud cases that are typically used to examine performance of boundary layer, microphysics, and shallow convective parameterizations, while cases such as ARM97, ARM95, TWP, and GATE are cases that can be used to evaluate shallow and deep convective parameterizations. The E3SM SCM library contains IOP files from more recent and modern cases, such as GOAMAZON, RACORO, and DYNAMO; many of these are unique to the E3SM SCM.

The E3SM SCM case library is publicly available on the E3SM SCM Github project wiki (https://github.com/E3SM-

Project/scmlib/wiki/E3SM-Single-Column-Model-Case-Library). The SCM user needs only to clone the github repository, which includes the scripts required to run the SCM cases. Note that the code needed to run the E3SM SCM is included with the standard E3SM release code. The user then needs to modify the header of the script for the desired case they wish to perform and then execute the script, which will compile and run the SCM for the desired case. We chose to provide and maintain separate scripts for each particular case, with the unique settings, switches, and idealizations for each case set in the script. An

alternative approach is to provide the user with a universal script that can be used to run all cases and to hardcode each case into the E3SM infrastructure as a particular run type (known as a "compset" in the CAM/E3SM parlance). We find that providing unique scripts for each case provides more transparency, while the details of "compsets" tend to remain under-the-hood to most E3SM users. Our approach also provides the user with more flexibility to switch on/off specific idealizations or settings, allowing them to perform sensitivity studies.

## 4    E3SM SCM Replay Option


A major advantage of the SCM using the same dynamical core as the full GCM is the ability to easily "replay" a single GCM column with a high degree of accuracy. This is a powerful tool where the user generates IOP forcing from a full E3SM run, with the intention to replicate a column of interest in SCM mode. This can be used to help diagnose model crashes due to unstable physics parameterizations, or to target and address chronic model biases in an efficient manner. It can also help to fill in the gap

for a particular regime or location where there is no forcing provided by the E3SM SCM library. The inherited SCM, which





used the Eulerian dynamical core would require the user to either generate the GCM forcing using the Eulerian dynamical core, which may not provide a faithful representation of the E3SM model and was technically challenging to do because the Eulerian dynamical core wasn't supported for GCM runs, or to modify the forcing data to allow the Eulerian dynamical core SCM mode to use it (as documented in Gettelman et al. 2019). Since the E3SM SCM uses the same dynamical core as the full

GCM, the method to replay a GCM column is straightforward and accurate.

Though the E3SM SCM Replay option is accurate, it cannot provide a fully bit-for-bit representation of a GCM column. This is because the GCM and SCM will only give bit-for-bit answers if they do exactly the same calculations. In GCM mode, the end of dynamics state is computed via a series of sub-stepped loops. For the SCM, the net effect of these loops must be encapsulated by the end-of-step values minus the beginning of step values, divided by the timestep. This tendency is then

added to the SCM state using forward Euler timestepping. Since the GCM and SCM calculations are not identical, roundoff level differences occur. This issue could in principle be resolved using quadruple precision output but we found the related difficulties associated with this to not be worth a roundoff level gain. Our approximate method has proven suitable for most scientific applications of interest to E3SM users. In section 5.5, we demonstrate an example of using the Replay option.

## 5 Applications of the E3SM SCM

In this section we will demonstrate that the SCM can serve as a tool to reproduce and explore climatological biases within the E3SM model. We will also show an example of when the SCM cannot be used as a proxy for the full model. Finally, we will show an example of using the E3SM Replay option. Some major biases in the E3SM model include (but are not limited to) an overestimate of clouds in the Arctic, lack of subtropical maritime stratocumulus, lack of high clouds in the Tropical West Pacific (TWP) warm pool, timing of precipitation in the tropics and mid-latitudes, and a lack of precipitation over the Amazon

rainforest (Xie et al. 2018; Zhang et al. 2018; Golaz et al. 2018; Rasch et al. 2019). In this section we will attempt to replicate a select number of these biases with the SCM.

Unless otherwise stated, the SCM results presented in this paper use the short-term hindcast approach (Ma et al., 2015). The SCM is initiated every day at 00Z and run for two days, with prescribed large-scale forcing, surface turbulent fluxes and no nudging. The 24 to 48 hour forecasts in each simulation are then combined as a continuous timeseries. With the hindcast ap-

proach, the model is well constrained by the large-scale condition, allowing us to isolate problems related to parameterizations. It also avoids the possible impacts of nudging to the clouds and precipitation (Ghan et al. 2000; Randall and Cripe 1999; Zhang et al. 2014). We will, however, explore the differences between nudging and the short-hindcast mode in one example.

### 5.1 Diurnal Cycle of Continental Precipitation

As already stated, SCMs are a useful tool to explore biases due to the model's physical parameterizations. But are there certain

conditions and regimes under which the SCM is a better or worse proxy for the full GCM? While it has been demonstrated many times in literature (e.g. Golaz et al. 2002; Bogenschutz and Krueger 2013; Suselj et al. 2013) that boundary layer cloud cases (such as DYCOMS for stratocumulus and BOMEX for shallow cumulus, as an example) can serve as a useful surrogate





to explore and improve biases in the global model (due to the important cloud forming processes in these regimes being mostly locally driven), the question of whether precipitation due to deep convective processes can be replicated faithfully in SCMs is

less understood. Here we will attempt to replicate E3SM's biases in precipitation, both in the mean state and variability sense, to see when the E3SM SCM may be useful to exploit and investigate these biases.

  The diurnal cycle of precipitation, especially over land, is a mode of climate variability that GCMs have long struggled to simulate adequately (Covey et al. 2016; Lee et al. 2007). Over land the late afternoon peak of precipitation is typically associated with the transition of shallow to deep convection while the nocturnal peak is mostly due to elevated convective

systems associated with eastward propagating mesoscale convective systems. Many studies have attributed the GCMs inability to represent the diurnal cycle of precipitation to deficiencies in the moist convective parameterizations (Dia and Trenberth 2004; Lee et al. 2008), where model errors over land are associated with unrealistic strong coupling of convection to the surface heating (Lee et al. 2007; Xie et al. 2002). Thus, precipitation peaks in the model tend to occur too early over land during the day, especially in summer.

E3SMv1 strongly exhibits these aforementioned biases (Xie et al. 2019; hereafter XIE2019), especially when focused over the continental United States (CONUS, figure 9 of XIE2019). In the central plains of the US, observed precipitation peaks in the late evening time, whereas E3SM precipitation peaks around noon. Can E3SM SCM reproduce this bias and can we use the SCM to implement modifications to the parameterized physics that would help improve this longstanding issue? For this experiment we use version 2 of continuous forcing from the Southern Great Plains (SGP) ARM site (Tang et al. 2019; Xie

et al. 2004) that spans from 2004 to 2015, however for this study we only consider the warm season from 1 May through 31 August of each year. Note that multi-year SCM forcing allows us to perform robust statistical analysis rather than relying on a single case study as typically done in the past with SCM runs.

  To see if we can improve this bias in the SCM, we implemented a revised convective triggering function, as described in XIE2019, which has been shown to greatly improve the diurnal cycle of precipitation in E3SM simulations. This new

convective triggering is a combination of two methods, known as dynamic Convective Available Potential Energy (dCAPE) and the Unrestricted Launch Level (ULL).

  The top row of figure 1 displays the composite of the total precipitation from the periods sampled at the SGP site. While observations show a minimum of precipitation around noon, this is when E3SM SCM shows a maximum precipitation rate. This is representative of the bias found in E3SM simulations for a similar location over the North American Plains subset

region in XIE2019, where precipitation was tied a bit too closely to solar insolation and the nocturnal peak of precipitation was not represented. XIE2019 also found that after implementing the revised dCAPE and ULL triggering method the precipitation maximum was shifted to the nocturnal hours (figure 13b of XIE2019). Clearly, not only can the E3SM SCM replicate the original bias found in the global model, but the improved representation due to the new convective triggering is also depicted in our SCM experiments.

Due to the fact that the SCM can replicate the behaviors seen in the global model for this situation, we can further use this SCM case to explore the exact reason for this behavior. The bottom row of figure 1 conditionally samples our dataset for days when the observed precipitation predominately happens in the afternoon and nighttime. We segregate the days with afternoon





maximum precipitation by subsetting to days when the observed precipitation has a peak greater than 1 mm/day between 1300 to 2000 LST and when the peak rain rate is 1.5 times greater than any rain rate outside of 1300 to 2000 LST. The nighttime

precipitation days are classified as when the rain peak is greater than 1 mm/day with a peak time between 0000 to 0700 LST. From this analysis, it is clear that the largest impacts from the improved triggering in terms of precipitation timing occur on days when there is a nocturnal peak of precipitation, which the default E3SM model was missing. The combination of the dCAPE trigger, which prevents the convection scheme from activating too early in the afternoon, and the ULL method which improves the elevated nocturnal convection help to shift the precipitation to the night time hours, on days when it it observed.

Thus, this case makes an example of when the SCM can serve as a good proxy to replicate and improve GCM biases, as well as easily investigating under what scenarios an improved scheme is having the most impact.

## 5.2 Amazon Precipitation Bias

Another major bias in E3SM which is characteristic of most GCMs, is lack of precipitation over the Amazon (Fig 9 of Xie et al. 2018). E3SM has a climatological dry bias upwards of 4 mm/day in this area which, while not as severe as most GCMs, is

a longstanding bias that negatively impacts feedbacks to/from the land model. To see if we can replicate this bias in the E3SM SCM we use the Green Ocean Amazon (GOAMAZON) case (table 1), which is a two year campaign taking place around the urban region of Manaus in central Amazonia from 2014-2015.

The top panel of figure 2 displays the annual cycle of precipitation for the SCM, observations, and the column closest to the GOAMAZON point for the E3SM GCM. Similarly, the bottom panel of figure 2 displays a composite of the daily cycle

of precipitation. For this location, the radar derived precipitation rate has an annual mean of 6.56 mm/day, the SCM an annual mean of 6.98 mm/day, and the GCM 6.07 mm/day. Therefore, here we see an example where the SCM does not faithfully represent the bias exhibited by the GCM in terms of the climatological rate of precipitation.

The reasoning why the GCM Amazon dry bias cannot be replicated in the SCM is likely because this bias is primarily due to a misrepresentation of the large-scale environmental conditions in the GCM, rather than by parameterized deficiencies.

This is important information for E3SM developers and the analysis team. Figure 3 displays the observed composite large-scale vertical velocity, relative humidity, and winds at the GOAMAZON location and compares these to the GCM simulated variables. The largest discrepancies between GCM and observations occur during the boreal summer months, which correlates to the period of the most pronounced bias in the climatological rain rate in the GCM. The SCM is driven by observed large-scale forcing, thus is not subject to the errors that drives the GCM biases.

In addition, it is well understood that for deep convection precipitation is usually balanced mainly by advective moisture convergence, which is prescribed in these experiments. Therefore, this is a prime example of a situation where the SCM is not a useful tool to help improve GCM biases but does suggest that efforts should be spent on improving the large-scale circulation, or remote biases, which are probably responsible for the Amazon precipitation bias. The SCM, however, can replicate the bias related to the early onset of precipitation (similar to that seen in fig. 1), thus supporting the idea that the diurnal cycle involves

shorter timescales and therefore looks more like the free-running GCM solution than the observed values.





### 5.3 Arctic Clouds

Zhang et al. (2018) show that E3SM suffers from an overestimate of Arctic clouds, mostly in the form of too much liquid cloud. Here we use the Mixed-Phase Arctic Cloud Experiment (MPACE; Verlinde et al. 2007) case, which sampled clouds over open ocean near Barrow, AK, with the goal to collect observations to advance the understanding of the dynamics and microphysical

processes of mixed-phase clouds. This is a seventeen day case taking place in October 2004.

The top row of figure 4 displays the cloud fraction from observations and from the E3SM SCM. The bottom row shows the timeseries of the liquid water path (LWP) and ice water path (IWP) for observations (black curve) and the default E3SM SCM in hindcast mode (red curve). As found in the GCM, we see a general overestimate of the cloud fraction in the E3SM SCM and a tendency for the E3SM SCM to overestimate LWP. This is in agreement with Zhang et al. (2018), which shows a bias in

the low-level cloud amount (their figure 3) and a negative shortwave cloud radiative effect bias in the Arctic. As described in Caldwell et al. (2019) this behavior is related to mistakenly setting the efficiency of the Bergeron-Findeisen process, in which ice crystals grow through sublimation at the expense of supercooled water droplet to the very low value of 0.1 in the v1 release. To test the impact of this choice, we set the Bergeron efficiency to 1.0. The result is a dramatic decrease in the amount of cloud liquid mixing ratio (third row of fig. 4 and blue curve of bottom row). This example illustrates the ease with which the SCM

can be used to explore the impact of parametric assumptions. Note, however, that this quick SCM test may not always capture the sensitivity of the full GCM and our quick test doesn't account for needed retuning to compensate for altered Bergeron efficiency. Though, how E3SM simulations would respond in the climatological sense, and what degree of retuning would be necessary by adjusting this efficiency parameter, is something that the SCM cannot afford to offer.

### 5.4 Hindcast vs. Nudging

As previously mentioned, we chose to perform the majority of experiments in this paper in short-term hindcast mode. However, the E3SM SCM also comes with an option to nudge temperature and moisture to observed values. By default the E3SM SCM uses a nudging timescale of three-hours. It is interesting to note that the solution obtained for the MPACE case is strongly dependent on the technique used to constrain the mean state. The fourth row of fig. 4, which uses nudging, clearly shows a very different solution in terms of the cloud and ice mixing ratio when compared to the hindcast simulation on row two.

The simulation with nudging tends to produce less liquid cloud and virtually no ice. Randall and Cripe (1999) extensively discussed the nudging method for SCMs and conclude that the impact of nudging on SCM simulation depends on the model biases produced without nudging, thus there is no solid theory on what can be expected from a particular model while using nudging.

Figure 5 displays the timeseries of the observed temperature profiles for the MPACE period, in addition to the temperature

biases for the E3SM SCM runs using hindcast and nudging methods. Obviously, since the nudged run is continually forced towards observations, the temperature bias is near zero for the duration of the run. Conversely, the hindcast run allows the temperature biases to grow over each 48 hr run and are therefore likely to be more representative to the E3SM model bias and therefore provide a more faithful representation of the model. This begs the question; which method should be used for




E3SM SCM simulations? The answer likely depends on the goal of the particular user. If one simply wants to use the SCM as
a proxy for E3SM performance, to replicate GCM biases and provide potential fixes for these biases, then running the SCM in
short-term hindcast or free running mode (for short IOP cases) is likely the best option. This will allow the mean state model
biases to evolve, but not drift, in a manner similar to the GCM and will likely provide a more faithful representation in terms
of cloud representation.

If, however, one is using the SCM for the purposes of parameterization development/implementation and wishes to assess
their new parameterization in conditions with little to no mean state bias (e.g. to avoid compensating errors), then the nudging
method is likely preferable. For instance, the results seen using nudging vs. hindcast for MPACE clouds may suggest that
Arctic clouds simulated in E3SM are an artifact of compensating errors. When the observed temperature and moisture profiles
are used, we see the model struggles to produce any ice cloud at all, which is in conflict with observations. This suggests that
E3SM developers may need to reevaluate either the parameterizations and/or tuning choices in order to get a desirable solution
when the temperature and moisture most resemble observations. In addition, caution should be warranted when using nudging,
since constantly nudging to the observed temperature and moisture state inherently breaks the water and energy budget by
acting as artificial source. The sequentially splitting techniques that E3SM uses could in theory be obscuring the direct effects
of this, which could be leading to the artificial reduction of condensate. Though, this idea needs to be explored more.

### 5.5 Example of Using the E3SM SCM Replay Option

Xie et al. (2018), Golaz et al. (2018), and Zhang et al. (2018) all report substantial bias of high clouds in the Tropical West
Pacific (TWP). Figure 6 displays the difference of E3SM simulated high clouds versus observations and shows a climatological
negative bias upwards of 40 percent in this region. This is one of the most severe cloud biases in the model and it would be
useful to investigate the cause of this bias in the context of the SCM. However, the E3SM case library does not have forcing
at the location of the heart of this bias. IOP forcing such as TWP-ICE (location indicated by an open red star in figure 6) is
located at the edge of this bias where the model has a good representation of high clouds. Therefore, this is an instance where
the Replay mode can help us.

We wish to replay a column near the location where the bias is most severe. Therefore we choose a location near 5°N and
140°E (see yellow star in figure 6). The bias in this location is most prevalent during the boreal summer months, therefore we
chose August as the month we will replay in SCM mode. In our experimental setup we simply run the GCM with climatolog-
ically prescribed SSTs for a year (starting in January) by configuring the simulation with a single directive ("-e3sm_replay"),
which will generate the appropriate forcing to replay a column at every E3SM timestep. To reduce the amount of output gen-
erated, we choose to do a regional subset of the forcing (instructions for this provided at the E3SM SCM wiki). We also chose
to output initial condition files at the start of every month so that our SCM can start from the same state as the GCM.

Once the simulation is over we use scripts provided in the E3SM case library to replay our column of choice. The inputs we
need to specify are the E3SM generated forcing file, initial condition file, the latitude and longitude we wish to simulate, as
well the desired start date and run duration.





Figure 7 displays the monthly mean profiles of cloud fraction, cloud liquid mixing ratio, and cloud ice mixing ratio for the SCM and GCM run for the column of interest. Observational guidance is provided from CALIPSO, CloudSat, and Moderate Resolution Imaging Spectroradiometer (MODIS) in a merged product called C3M (Kato et al. 2010). The GCM and SCM

profiles are averaged over August of the first year of the simulation performed with climatological SSTs, while the C3M data represents the average of August 2006-2010. Figure 7 clearly shows that E3SM underestimates cloud, not only at the upper-levels but also the lower and mid levels by a substantial amount for this column. While cloud liquid mixing ratio is represented with somewhat reasonable magnitude by the GCM, cloud ice is substantially under predicted. Thus the combination of the low cloud fraction and cloud ice is likely driving the radiation biases seen in the full GCM for this region.

From figure 7, it is also clear that the SCM Replay mode is also very capable of reproducing the full GCM, as cloud profiles exhibit nearly identical behavior. As a reminder, fully bit-for-bit results are not expected with the E3SM Replay mode due to the fact that the dynamics tendency calculation is applied differently than in the full GCM. However, we show here that the Replay mode can faithfully represent the behavior of the GCM. While the Replay mode cannot provide information on whether the warm pool cloud bias is due to parameterization deficiencies or discrepancies in the large-scale, we can use the

SCM Replay method to perturb parameterization tunable parameters in an efficient way to explore the effect they might have on the high cloud bias.

An example of this, we run the SCM in Replay mode for this column but with the Bergeron efficiency set to 1.0 (blue dashed curve in figure 7), as in section 5.3. In this experiment, while we see noticeable effects in the mid-troposphere in terms of the reduction of cloud liquid, there is little effect towards the increase of cloud fraction or cloud ice mixing ratio. Simultaneously,

we also performed several experiments where we perturbed the critical thresholds of the relative humidity for the ice cloud fraction closure (Gettelman et al. 2010), but we saw no noticeable changes in the simulation of the cloud profiles (not shown). While these experiments were not successful towards improving this bias in E3SM, they allowed us to efficiently rule out potential culprits in the tuning choices while avoiding wasting computational resources of testing the same experiments in long climate integrations.

**6 Summary and Discussion**

This paper describes the E3SM SCM, including modifications made to it since we adopted it from CAM, and how this configuration can be useful for model development and evaluation. A number of important upgrades were made to E3SM SCM since the inherited version, including the ability for the user to specify how the aerosols are treated to avoid unscientific case set ups due to the fact that E3SM initializes all aerosol concentrations to zero. Idealizations have also been implemented and turned on

by default, depending on the IOP forcing the user selects, to ensure an apples-to-apples comparison with LES benchmarks that the IOP forcing was meant to replicate. Most importantly, the E3SM SCM is now configured to work with the same dynamical core as the full GCM. This ensures that the SCM runs with the same large-scale vertical advection scheme, time step, and physics-dynamics coupling methods as the full model. It also allows the user to trivially "replay" a column of the full GCM with the SCM without the need to interpolate initial condition files or forcing files from one dynamical core to the next.





The E3SM SCM also has an extensive library of IOP cases that span the traditionally used GCSS boundary layer cloud cases (i.e. BOMEX, DYCOMS, RICO) and standard deep convection cases (i.e. ARM97, GATE). We also include IOP forcing files from more recent and modern cases, such as GOAMAZON, RACORO, and DYNAMO; many of these which are unique to the E3SM SCM. For example, the E3SM can simulate conditions at ARM SGP for twelve continuous years. This allows for robust GCM-like statistics to be generated in a computationally efficient manner. Scripts to run each individual case are available at

https://github.com/E3SM-Project/scmlib/wiki/E3SM-Single-Column-Model-Case-Library and many have been scientifically validated. The user need only supply paths to relevant output directories if running on E3SM support machines.

We provide some examples of when the E3SM SCM may prove to be a useful proxy for GCM performance. For instance, we are able to successfully replicate the diurnal cycle of precipitation bias in the GCM by using forcing generated at ARM SGP. This bias is mostly due to deficiencies in the triggering mechanism in the convective parameterization that is unable to properly

handle elevated convection. By implementing the trigger improvements documented in XIE2019, we are able to reproduce the same improved diurnal cycle of precipitation in the SCM found in global simulations. However, we were unable to replicate the seasonal cycle of dry Amazon bias with the SCM. We conclude that the root cause of the bias is due to improper representation of the large-scale environment rather than a deficiency with the parameterizations.

Using Arctic clouds as an example, we use the SCM to experiment with tunable parameter changes to evaluate the sensitivity

of the high latitude cloud bias. We report positive effects with the tuning of one parameter for this particular regime, but we caution that the SCM cannot inform how a full GCM simulation and radiation balance would be impacted with a modification. We also compare the SCM in hindcast or free running mode versus a run where the SCM is nudged to observations. By running in hindcast or free-running mode the SCM allows the model biases in temperature and moisture to naturally develop which gives a better proxy with the model behavior in the full GCM and therefore should be used if trying to replicate E3SM behavior.

Nudging the SCM to observations may not provide a proxy with the full GCM and the behavior that could deviate significantly from E3SM global runs. This mode is, however, potentially useful if trying to improve or implement a parameterization while avoiding compensating errors. We caution the user on the potential unintended consequences of adding artificial sources that nudging could introduce.

We also demonstrate that the Replay mode in the E3SM SCM can faithfully replicate a column of the GCM, though bit-for-

bit replication is not possible in the current implementation. This mode is useful when trying to simulate a particular regime or region that the extensive E3SM case library does not cover. In our example, we replicated the high cloud bias in the Tropical Pacific Warm Pool. While the SCM cannot inform us directly whether biases are caused predominately by deficiencies in the model physics or the large-scale flow; it can provide clues about the culprit. This allows model developers to focus their energies more efficiently towards a solution.

The E3SM SCM is mature and should be a first step in the model physics development and implementation process. With the extensive case library and the ability to simulate many different regimes, users can gain valuable insights on their development efforts and efficiently fix bugs. The SCM is also an important tool for addressing long standing biases in the model; its incredible efficiency makes large sets of perturbed parameter tests easy. In addition, model instabilities that may arise in the full GCM



can be investigated efficiently using the easy to use SCM Replay mode, which is a powerful tool that can faithfully replicate a

column of the full GCM.

*Code and data availability.*   The model code used in this study is located at https://doi.org/10.5281/zenodo.3742207, while the scripts used to generate the SCM simulations in this paper are located at https://github.com/E3SM-Project/scmlib/wiki/E3SM-Single-Column-Model-Case-Library, and the output from the SCM hindcast simulations can be found at https://portal.nersc.gov/project/mp193/sqtang/E3SM_SCM_runs/.

*Author contributions.*   LLNL conceived the study and was the primary developer of the SCM. Brookhaven National Laboratory did some of

the preliminary development work of the SCM, while NOAA provided some crucial code contributions for the Replay mode for the SCM. All authors contributed to interpreting the results.

*Competing interests.*   The authors declare that they have no competing interests

*Acknowledgements.*   This work was part of the CMDV Software Modernization program, funded by the U.S. Department of Energy (DOE) Office of Biological and Environmental Research. This work was performed under the auspices of the U.S. Department of Energy by LLNL

under contract DE-AC52-07NA27344. LLNL IM: LLNL-JRNL-802301-DRAFT.



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



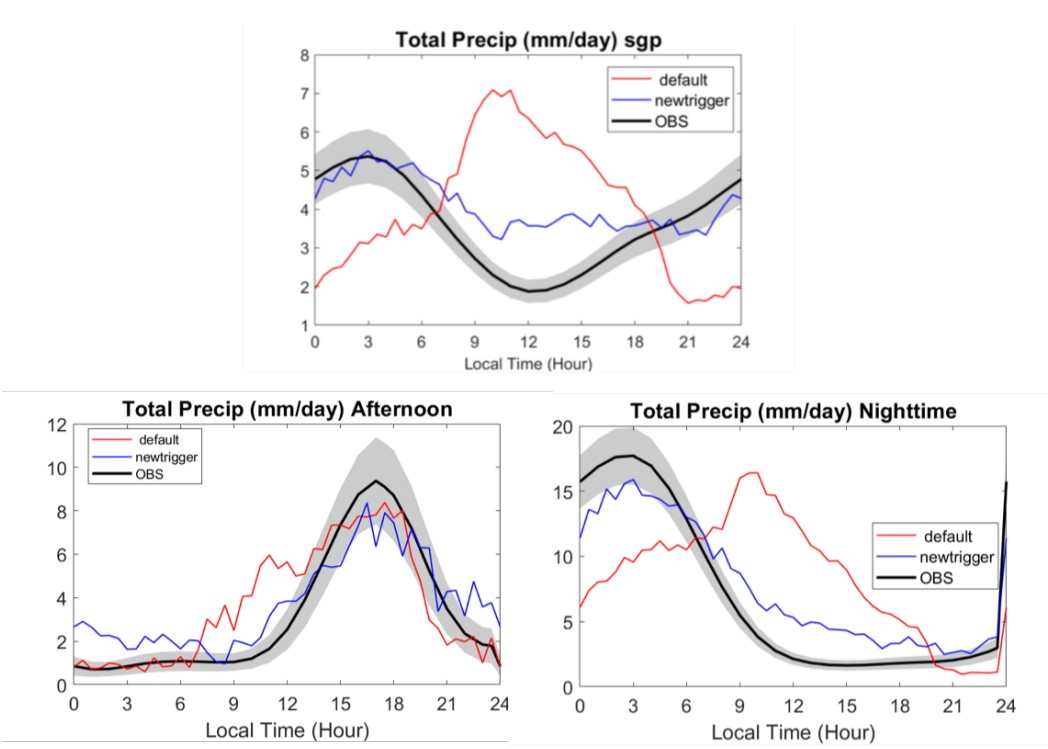

**Figure 1.** Composite of the total precipitation (convective + large scale) in local time for the E3SM SCM (red curve), E3SM SCM with the convective modifications documented in (Xie et al. 2019; blue curve), and observations (black curve) from the Southern Great Plains (SGP) ARM site version 2. Top panel depicts all time samples from 1 May through 31 August from 2004 to 2015. Bottom left panel represents periods when the observed precipitation has a peak greater than 1 mm/day between 1300 to 2000 LST and when the peak rain rate is 1.5 times greater than any rain rate outside of 1300 to 2000 LST. The Bottom right panel represents periods when the observed precipitation is greater than 1 mm/day with a peak time between 0000 to 0700 LST.



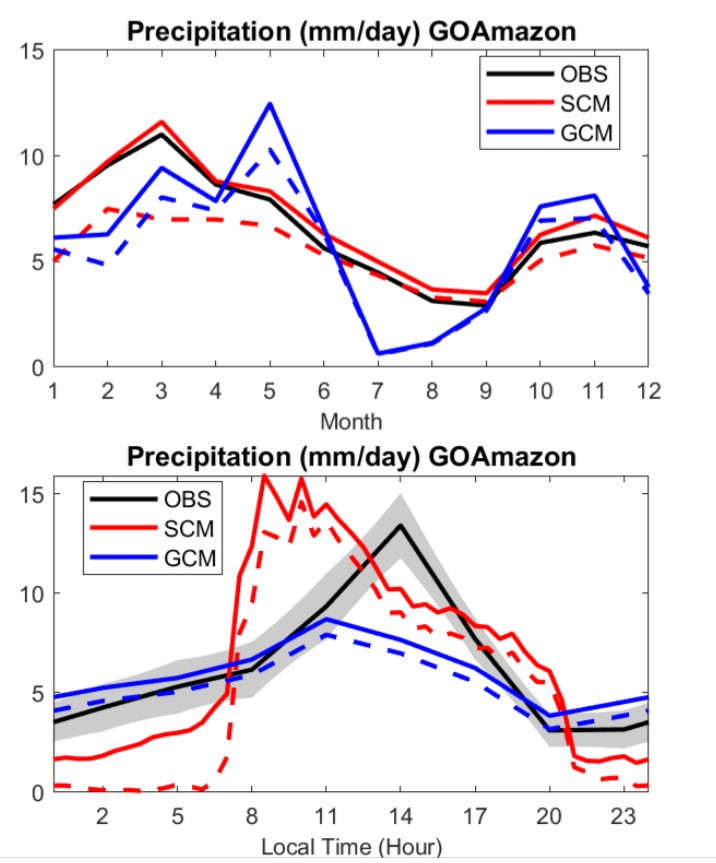

**Figure 2.** Precipitation from the GOAMAZON field campaign for SCM (red curve) and observations (black curve). The GCM results (blue curve) are taken from an E3SM run in the column closest to the GOAMAZON location (3°S and 300°E) from January through December of 2014. Top panel represents the annual cycle of precipitation, while the bottom panel represents the daily cycle. Solid curves represent the total precipitation rate, while dotted curves represent the contribution of the convective precipitation.

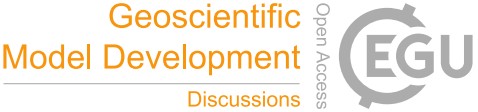

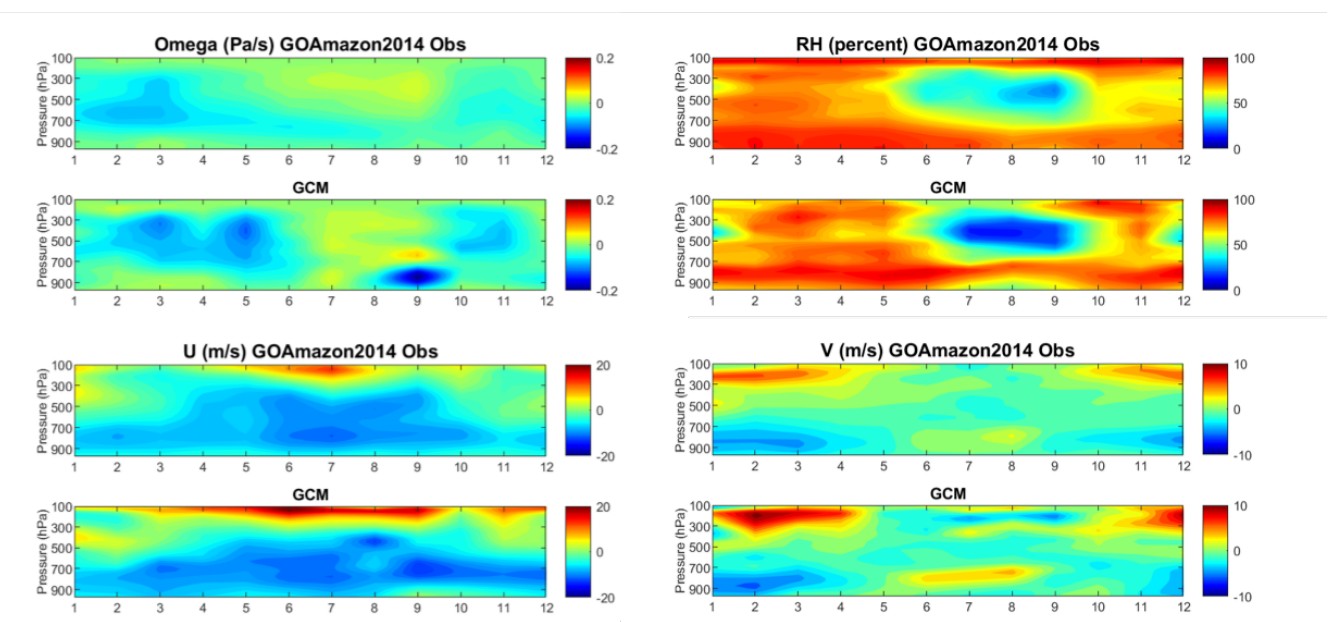

**Figure 3.** Annual cycle of observed versus E3SM simulated environmental states for the GOAMAZON location for large-scale vertical velocity (top two left panels), relative humidity (top two right panels), zonal wind (bottom two left panels) and meridional wind (bottom two right panels).





**Figure 4.** Top panel displays the evolution of the vertical cloud structure for cloud fraction (observations on the left and E3SM SCM on the right) for the MPACE case from October 2004. The left panel of rows three through four represent the cloud liquid mixing ratio while the right panels of rows three through four represent cloud ice mixing ratio from E3SM SCM simulations). The second row represents simulations using the hindcast method, the third row represents simulations with the Bergeron-Findeisen process set to the default tuning value, and row four represent the simulations where temperature and moisture are nudged to observations. The bottom row displays the evolution of the integrated liquid (left) and ice (right) path for the various configurations mentioned.





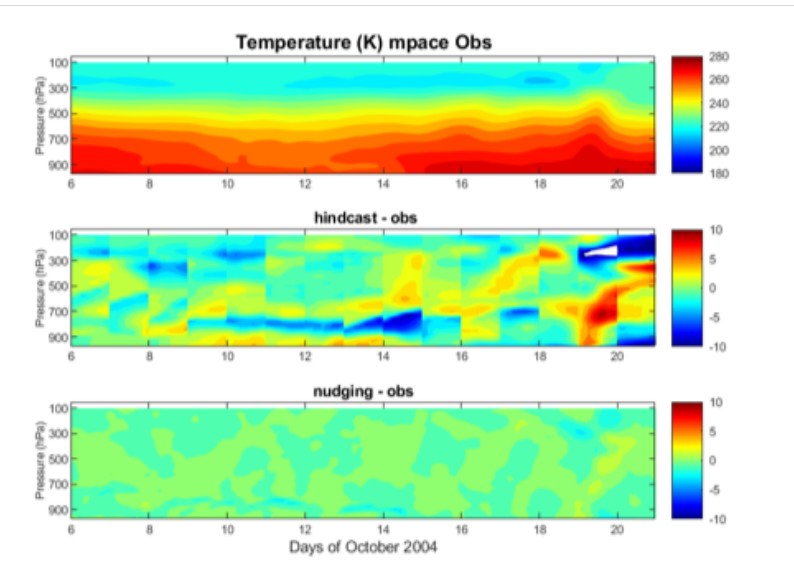

**Figure 5.** Vertical evolution of temperature from observations (top row) for the period during the MPACE field campaign. Also displayed are the temperature biases for the E3SM SCM run in hindcast mode (middle row) and for E3SM SCM run with nudging (bottom row).

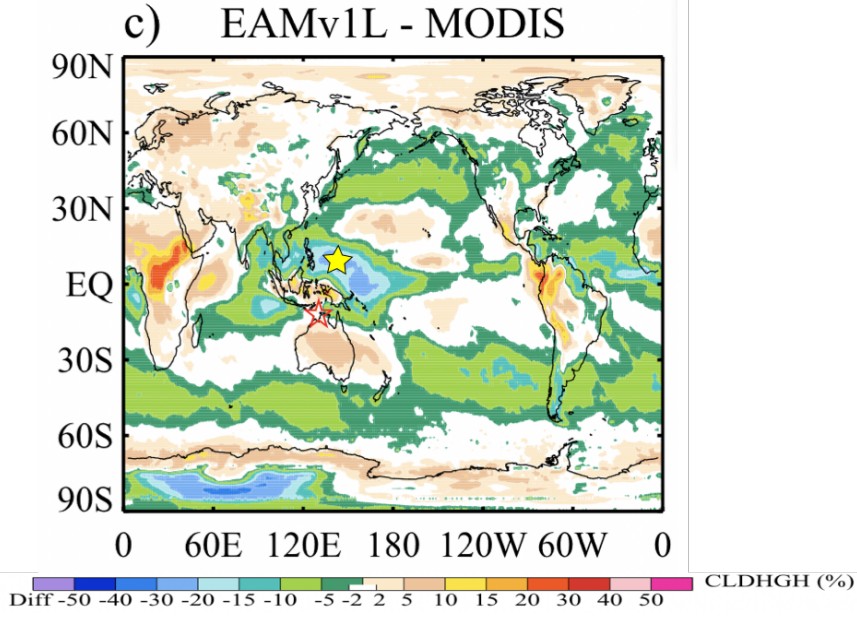

**Figure 6.** Climatological E3SMv1 high cloud bias, computed relative to MODIS observations, from Xie et al. (2018). Open red star shows location of the TWP field campaign, while solid yellow star shows the location we use for our E3SM SCM Replay experiment.



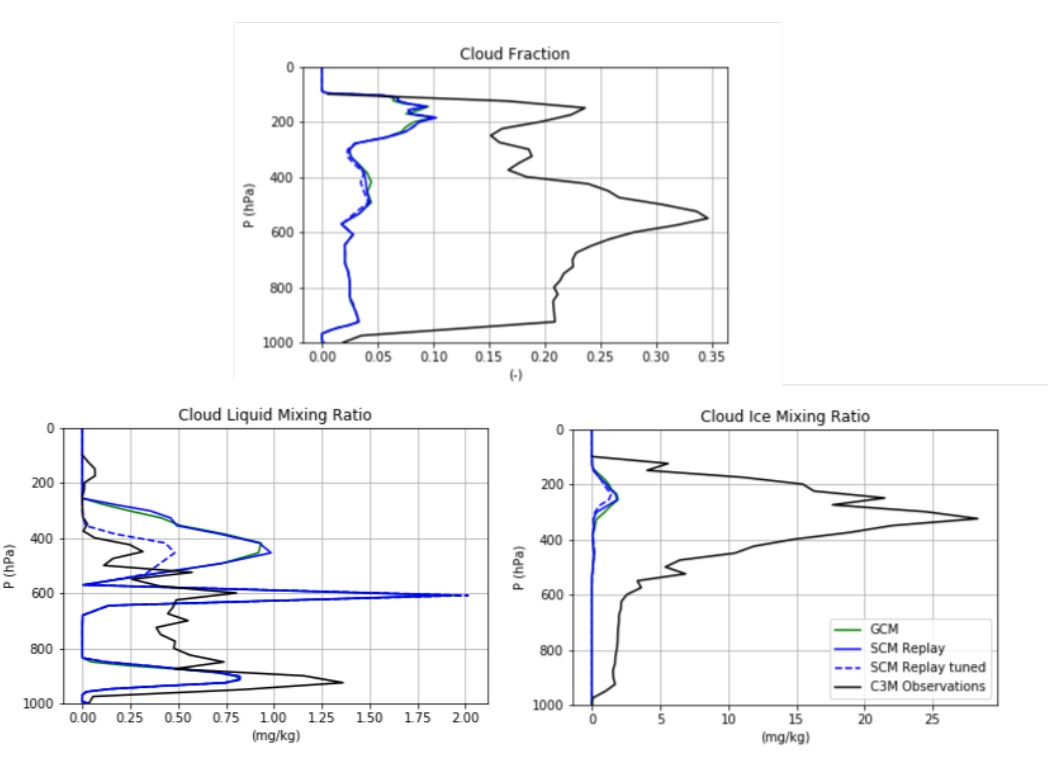

**Figure 7.** Temporally averaged profiles of cloud fraction (top panel), cloud liquid mixing ratio (bottom left panel), and cloud ice mixing ratio (bottom right panel) for C3M observations (black curves), GCM (green curves), and E3SM SCM Replay (blue curves) from location 5°N and 140°E. Profiles from observations represent average over August 2006-2010, while GCM and SCM profiles represent averages from August of a one year simulation using prescribed climatological SSTs.





**Table 1.** E3SM Case Library (part one)

| Name | Long Name | Lat | Lon | Start Date | Length | Reference | Regime |
|---|---|---|---|---|---|---|---|
| AEROSOL INDIRECT | Aerosol Indirect Effects in China | 32 | 117 | Nov 2008 | 28d | Liu et al. (2011) | Continental aerosols |
| ARM95 | ARM Southern Great Plains | 36 | 263 | Jul 1995 | 18d | Zhang et al. (1997) | Continental convection |
| ARM97 | ARM Southern Great Plains | 36 | 263 | Jun 1997 | 30d | Zhang et al. (1997) | Continental convection |
| ARM SGP | ARM Southern Great Plains | 36 | 263 | Jan 2004 | 12y | Xie et al. (2004) Tang et al. (2019) | Continental convection |
| ARM Shallow | ARM Southern Great Plains | 36 | 263 | Mar 2000 | 14h | Brown et al. (2002) | Continental shallow cumulus |
| ATEX | Atlantic Trade Wind Exp | 15 | 345 | Feb 1969 | 2d | Stevens et al. (2001) | Shallow cumulus |
| BOMEX | Barbados Ocean and Met Exp | 15 | 300 | Jun 1969 | 5d | Siebesma et al. (2003) | Shallow cumulus |
| DARWIN | ARM TWP ocean site | -12 | 131 | Oct 2004 | 5m | May et al. (2008) | Tropical convection |
| DYCOMSRF01 | Dynamics of Marine Stratocumulus | 32 | 239 | Jul 2001 | 2d | Stevens et al. (2005) | Stratocumulus |
| DYCOMSRF02 | Dynamics of Marine Stratocumulus | 32 | 239 | Jul 2001 | 2d | Ackerman et al. (2009) | Stratocumulus |
| DYNAMO AMIE | Dynamics of the Madden Julian Oscillation | -1 | 73 | Oct 2011 | 90d | Yoneyama et al. (2013) | Tropical convection |
| DYNAMO North Sounding | Dynamics of the Madden Julian Oscillation | 3 | 76 | Oct 2011 | 90d | Yoneyama et al. (2013) | Tropical convection |

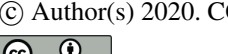



**Table 2.** E3SM Case Library (part two)

| Name | Long Name | Lat | Lon | Start Date | Length | Reference | Regime |
|---|---|---|---|---|---|---|---|
| GATEIII | GATE Phase III | 9 | 336 | Aug 1974 | 20d | Houze and Betts (1981) | Tropical convection |
| GOAMAZON | Green Ocean Amazon | -3 | 300 | Jan 2014 | 2y | Martin et al. (2016) Tang et al. (2016) | Tropical continental convection |
| ISDAC | Indirect and Semi-Direct Aerosol Campaign | 71 | 156 | Apr 2008 | 29d | McFarquhar et al. (2011) | Arctic clouds and aerosols |
| MC3E | Midlatitude Cont. Convective Clouds Experiment | 36 | 263 | Apr 2011 | 45d | Xie et al. (2014) Jensen et al. (2015) | Midlatitude convection |
| MPACE | Mixed Phase Arctic Clouds Exp | 71 | 206 | Oct 2004 | 17d | Verlinde et al. (2007) Xie et al. (2006) | Arctic clouds |
| MPACE-B | Mixed Phase Arctic Clouds Exp | 71 | 206 | Oct 2004 | 12h | Klein et al. (2009) | Arctic clouds |
| RICO | Rain and Cumulus over Oceans | 18 | 299 | Dec 2004 | 3d | Rauber et al. (2007) | Shallow cumulus |
| RACORO | Clouds with Low Liquid Water Depths Optical Radiative Obs | 36 | 263 | May 2009 | 26d | Vogelmann et al. (2012) | Continental low clodus |
| SPARTICUS | Small Particles in Cirrus | 37 | 263 | Apr 2010 | 30d | Mace et al. (2009) | Cirrus |
| TWP06 | Tropical W. Pac. Conv. | -12 | 131 | Jan 2006 | 26d | May et al. (2008) Xie et al. (2010) | Tropical convection |
| TOGAIII | Tropical Ocean Global Atm. | -2 | 154 | Dec 1992 | 21d | Webster and Lukas (1992) | Tropical convection |