# Peer review of "The E3SM version 1 Single Column Model"

_Geoscientific Model Development, 2020_

## Referee Comment (RC1) · Anonymous Referee #1 · 15 Jun 2020

The discussion paper by Bogenschutz et al. describes single-column model (SCM) of the E3SM, version 1 model. In my opinion, this discussion paper is very suitable for GMD, is well written and deserves publication as a full paper. Below, I have a few comments that I think the authors should address:

1. Could the authors briefly explain how the surface conditions are prescribed for the SCM, and describe the options that are available for the surface conditions?

2. Is there a plan to develop an ocean-atmosphere SCM for studying coupled atmosphere-ocean processes (e.g. Hartung et al., 2018), considering that the E3SM includes these components? Do authors have an opinion if such coupled SCM would be useful from the point of parameterization development?

[Figure]

3. I really like a study of precipitation bias over the SGP and AMAZON sites and discussions about the representativeness of SCM model results for the three-dimensional model. Is there a way to predict this representativeness before conducting SCM experiments? I would naively think that comparing the dynamical and physical tendencies from the three-dimensional model could be a way to do this.

4. I think recent work by Smalley et al. (2019) on the SCM development and its use for parameterization testing and development deserves to be mentioned.

References:

- Hartung et al., 2018. An EC-Earth coupled atmosphere–ocean single-column model (AOSCM.v1_EC-Earth3) for studying coupled marine and polar processes. GMD. doi: 10.5194/gmd-11-4117-2018

- Smalley et al., 2019. A Novel Framework for Evaluating and Improving Parameterized Subtropical Marine Boundary Layer Cloudiness. MWR. doi: 10.1175/MWR-D-18-0394.1
* * *

---

## Referee Comment (RC2) · Anonymous Referee #2 · 17 Jun 2020

This paper summarizes updates, features, and new test cases for the SCM within the E3SM model ecosystem. The paper is well organized and generally well written. One highlight is the nice suite of test cases and reference to SCM features and when they should be used. This paper is worthy of publication once several issues are addressed below.

1. Section 2.3

Line 103, for the current dynamical core for CESM, cite a paper or code documentation. Lines 105-7, the SCM provides the large-scale vertical transport or the full dynamical core does? Are you saying the Eulerian and SE dynamical cores calculate vertical transport differently? Maybe it's the phrase "dynamical core in the SCM" that is confusing because that doesn't make sense. Please make clear what the full 3D dynamical

core calculates and what the SCM calculates and why that provides inconsistencies. Some explanation of how the codes are connected would be useful before delving into the details.

Line 109, forward in time is not specific enough. I think you should specify the scheme with an explanation as to why the differences in the dynamical core time integration matters for the SCM. For example, I can imagine that a staged explicit scheme will not mesh well with the leapfrog scheme because then the time step stages are at different times. In the same vein, why does the SCM have to use what the 3D dynamical core uses? Do they only share info at the outer final time step from a staged method?

Two paragraphs starting on line 112, the explanation of the SE grid needs more clarity. You need to explain how quadrilateral elements make up a sphere - a cube of faces that are then mapped to a sphere. You state that the SE grid must be initialized with a minimum resolution of points, I assume this is 1 per cube face? Do not use the HOMME/CAM-SE developers nomenclature of "ne4" unless you explain or cite. Then you can explain how you instantiate a low resolution version of SE and the SCM is only computed for one column (from one GLL point or for a point from the whole element?).

Rest of section 2.3, the advantages are just tacked on here. This should perhaps go somewhere else.

2. Lines 150-154, this sentence is repetitive and not clear. The SCM used the Eulerian dynamical core before you upgraded it, ok. So you are just saying that the SCM required files in the same Eulerian format or another strategy that is unclear. What does "it" at the end of the sentence refer to?

3. End of section 4: I was hoping for some idea of what was done to make SCM work within CAM-SE. Just code bug fixes? A bunch of little things? Were any scientific changes made?

4. Several paragraphs starting on Line 233, I am on board for annual bias not showing

up for SCM, but the daily precipitation bias for SCM is worse and different than GCM and obs. The meaning of this is implied in the next paragraph, line 249, that SCM can replicate early onset precipitation… but in a different thread of discussion, so that connection should be made more clear. Line 244, I don't see that, so you will need to plot bias as well. The statement "largest discrepancies occur during most pronounced bias" is not useful. 5. Line 268, what do you mean by "cannot afford to offer?" Do you mean it would take a long time to develop or it would make the SCM too expensive? Or both?

Minor comments: Will SCM change within version 2 of the E3SM? The use of "version 1" in the title implies that it is version dependent.

One minor exception to the good writing is the use of "which" and "that" (e.g. line 28,98,126, 225, 228, there may be more?) throughout and the use of tense (e.g. line 49, the paper will be organized, lines 111- 112 was then is).

Lines 42-44 and 56-58 are virtually ID, remove/adjust one of them. It's awkward where placed in the second instance.

Line 95 "The idealization switches added to the E3SM SCM framework includes" should use "include"

Line 98: Just say that some cases have idealizations turned on by default and move mention of specific cases later in the paper when they have been defined and explained.

Line 147 "high degree" of accuracy is imprecise. Relative to what?

Line 174 first mention of nudging- should explain/define, and line 176 what are the impacts of nudging?

Line 179 Where was this already stated?

Line 204, did Xie19 implement or just describe this revised function? It is not clear from

this description.

Line 255, once you have a fix you can see why the fix works. But what if you see a bias? How does it help you narrow down the source?

Line 298, Though -> Although

Line 318, what do you mean by observational guidance?

Line 332, "An example of this" should be "as an example of this"

Figure 4 caption: second sentence, you mean the 2nd through 4th rows?

———————————————————

---

## Author Comment (AC2) · 16 Jul 2020

The authors thank reviewer 2 for taking the time to provide us with a careful review of our paper and for the positive feedback and helpful suggestions. Please see our responses below, that are reflected in our revised document and track changes.

1. Section 2.3 Line 103, for the current dynamical core for CESM, cite a paper or code documentation. Lines 105-7, the SCM provides the large-scale vertical transport or the full dynamical core does? Are you saying the Eulerian and SE dynamical cores calculate vertical transport differently? Maybe it's the phrase "dynamical core in the SCM" that is confus- ing because that doesn't make sense. Please make clear what the full 3D dynamical core calculates and what the SCM calculates and why that provides

inconsistencies. Some explanation of how the codes are connected would be useful before delving into the details.

Author reply: For the CESM dynamical core reference, we refer the reader to Lin and Rood (1997) and to the CAM5 technical document of Neale et al. (2012) for further information how the dynamical core was implemented into CAM5.

We realize that the explanation of the dynamical core and how the large scale forcing was computed was confusing. We have reworded this with the following statements:

"This is a problem because while the horizontal advection fields are provided by the IOP forcing files, the dynamical core is still responsible for computing the large scale vertical transport if this is not prescribed in the forcing file. In the E3SM/CAM SCM the only part of the dynamical core that is exercised is the computation of the large-scale vertical transport. The Eulerian and SE dynamical cores use two different methods for this computation, the former uses a simple Eulerian calculation while the later uses a semi-Lagrangian method. Therefore, the inherited SCM was inconsistent with the full GCM in regards to how the large scale vertical advection was computed."

Referee comment: Line 109, forward in time is not specific enough. I think you should specify the scheme with an explanation as to why the differences in the dynamical core time integration matters for the SCM. For example, I can imagine that a staged explicit scheme will not mesh well with the leapfrog scheme because then the time step stages are at different times. In the same vein, why does the SCM have to use what the 3D dynamical core uses? Do they only share info at the outer final time step from a staged method?

Author reply: We changed "forward in time" to "the SE dynamical core uses a third-order five-stage explicit Runge-Kutta (RK) method as described in Dennis et al. (2012)". We note that the SCM technically does not have to use the same dynamical core as the full model, but doing so helps to minimize the differences between the full model and the SCM. In this case, because the Eulerian core uses a leapfrog method, the

physics time step has to be 2x the dynamics timestep. This is different than the full model in which the dynamics time step is equal to the physics timestep. Therefore, different timestep settings between the SCM and full 3D model may not provide a fully accurate representation of the full model in the SCM if certain schemes happen to be sensitive to timestep, as we note in section 2.3.

Referee comment: Two paragraphs starting on line 112, the explanation of the SE grid needs more clarity. You need to explain how quadrilateral elements make up a sphere - a cube of faces that are then mapped to a sphere. You state that the SE grid must be initialized with a minimum resolution of points, I assume this is 1 per cube face? Do not use the HOMME/CAM-SE developers nomenclature of "ne4" unless you explain or cite. Then you can explain how you instantiate a low resolution version of SE and the SCM is only computed for one column (from one GLL point or for a point from the whole element?).

Author reply: We have revised this paragraph to be more clear. It now reads as:

"Ideally, we want the SCM to be as close a proxy to the full GCM run as possible. Thus we upgraded the SCM dycore to use the same spectral-element (SE) dynamical core used by E3SM. Even though horizontal advection is prescribed in the SCM, the dycore still plays an important role for vertical advection. As described in Dennis et al. (2012), the SE dycore operates on quadrilateral elements whose Gauss-Lobatto-Legendre (GLL) quadrature points form the physics columns targeted by the SCM. Because there are many physics columns within each spectral element, it is impossible to initialize a single physics column when running a SE-dycore SCM. In this context the simplest way to initialize the SCM dycore is to "trick" the model by initializing the dynamical core using a low resolution global configuration, but then only actually use a single physics column for our calculations. We do this using the lowest-resolution configuration supported by E3SM, which contains 96 elements and corresponds to a grid spacing of approximately 7.5 degrees at the equator. Our strategy requires slightly more memory (to initialize the whole dynamics grid) but no more computational

expense than if we initialized just one column (because we only perform physics and vertical advection calculations on a single column)."

Referee comment: Rest of section 2.3, the advantages are just tacked on here. This should perhaps go somewhere else.

Author reply: We agree these lines were out of place, and we moved these lines to the Summary and Discussion section.

Referee comment: 2. Lines 150-154, this sentence is repetitive and not clear. The SCM used the Eulerian dynamical core before you upgraded it, ok. So you are just saying that the SCM re- quired files in the same Eulerian format or another strategy that is unclear. What does "it" at the end of the sentence refer to?

Author reply: We removed the repetitive statement and the ambiguous reference to "it". The statement was clarified to mean that when running the E3SM SCM Replay option does not require post processing of the forcing files as does CAM's version of this functionality. This is mostly because forcing terms are slightly different and reside on different grid structure between dynamical cores.

Referee comment: 3. End of section 4: I was hoping for some idea of what was done to make SCM work within CAM-SE. Just code bug fixes? A bunch of little things? Were any scientific changes made?

Author reply: The main "crux" to get this to work was training the SE dycore to run with one column, which has been described in the second paragraph of section 4. Beyond that, it was a simple matter of isolating the dynamical core so that the only calculation it returns was for the large-scale vertical advection (i.e. no horizontal advection). At the end of section four we add:

"This was the main challenge towards being able to use the SE dynamical core in the SCM setting, after which we trained the SE dynamical to only calculate the large-scale vertical advection (i.e. no horizontal advection) in the column of interest if in SCM

mode."

Referee Comment: 4. Several paragraphs starting on Line 233, I am on board for annual bias not showing up for SCM, but the daily precipitation bias for SCM is worse and different than GCM and obs. The meaning of this is implied in the next paragraph, line 249, that SCM can replicate early onset precipitation. . . but in a different thread of discussion, so that connection should be made more clear. Line 244, I don't see that, so you will need to plot bias as well. The statement "largest discrepancies occur during most pronounced bias" is not useful. 5. Line 268, what do you mean by "cannot afford to offer?" Do you mean it would take a long time to develop or it would make the SCM too expensive? Or both?

Author reply: We realize the confusion here likely relates to poor wording choices in our original document. Thus, starting on line 233 we have clarified things with different wording choices and we also brought the discussion of the early onset of precipitation earlier when we discuss the bias results. Also, our choice of "cannot afford to offer" was a very poor choice of words. What we meant to say was that the "SCM is unable to provide insights"on this matter. We have modified the text to reflect this.

Referee Comment: Minor comments: Will SCM change within version 2 of the E3SM? The use of "version 1" in the title implies that it is version dependent.

Author reply: No. GMD requires us to note which version of the model we refer to in the title. However, in the conclusions we now state that we do not expect the E3SMv1 infrastructure to change with E3SMv2.

Referee comment: One minor exception to the good writing is the use of "which" and "that" (e.g. line 28,98,126, 225, 228, there may be more?) throughout and the use of tense (e.g. line 49, the paper will be organized, lines 111- 112 was then is).

Author reply: Thank you. We feel we have addressed these issues.

Referee comment: Lines 42-44 and 56-58 are virtually ID, remove/adjust one of them.

It's awkward where placed in the second instance.

Author reply: This has been resolved by modifying the later statement.

Referee comment: Line 95 "The idealization switches added to the E3SM SCM framework includes" should use "include"

Author reply: Resolved.

Referee comment: Line 98: Just say that some cases have idealizations turned on by default and move mention of specific cases later in the paper when they have been defined and ex- plained.

Author reply: This has been resolved as suggested by the reviewer.

Referee comment: Line 147 "high degree" of accuracy is imprecise. Relative to what?

Author reply: This has been reworded to read "to replicate a specific GCM column with only round-off error differences"

Referee comment: Line 174 first mention of nudging- should explain/define, and line 176 what are the impacts of nudging?

Author reply: We have now defined nudging at first instance. We moved text from section 5.4 that discusses the impacts of nudging to this line, which feels more in place.

Referee comment: Line 179 Where was this already stated?

Author reply: We removed this statement to avoid the inconsistency.

Referee comment: Line 204, did Xie19 implement or just describe this revised function? It is not clear from this description.

Author reply: They implemented and analyzed the effects of the revised function. We changed the wording to make this more clear.

Referee comment: Line 255, once you have a fix you can see why the fix works. But what if you see a bias? How does it help you narrow down the source?

Author reply: It is unclear to us what the reviewer is referring to here. This line is describing the MPACE case meteorology and setup and has not mentioned anything about a bias. Unless further clarification can be provided, nothing has been changed here.

Referee Comment: Line 298, Though -> Although

Author reply: Fixed.

Referee comment: Line 318, what do you mean by observational guidance?

Author reply: Changed this to simply state "For observations we use…"

Referee comment: Line 332, "An example of this" should be "as an example of this"

Author reply: Fixed

Referee comment: Figure 4 caption: second sentence, you mean the 2nd through 4th rows?

Author reply: Yes, we have fixed this.

---

## Author Response (AR1)

The authors thank reviewer 1 for taking the time to provide us with a careful review of our paper and for the positive feedback and helpful suggestions. Please see our responses below, that are reflected in our revised document and track changes.

1. Could the authors briefly explain how the surface conditions are prescribed for the SCM, and describe the options that are available for the surface conditions?

We added a discussion on the surface fluxes at the beginning of section 2. The new text reads as:

Similar to SCAM, surface fluxes in E3SM can be prescribed, and is the default setting if this information is available in the case forcing file. Otherwise, the surface fluxes are computed interactively via the land model or the data ocean model, using prescribed sea surface temperatures.

2. Is there a plan to develop an ocean-atmosphere SCM for studying coupled atmosphereocean processes (e.g. Hartung et al., 2018), considering that the E3SM includes these components? Do authors have an opinion if such coupled SCM would be useful from the point of parameterization development?

There is currently no such plan for this extension in the E3SM SCM. However, we cite Hartung et al. 2018 and mention the importance that this functionality could provide with the following statement at the beginning of section 2:

The E3SM SCM does not currently support running an interactive ocean model, such as the work presented in Hartung et al. (2018), that may be a useful framework towards understanding parameterization feedbacks and climate sensitivity.

3. I really like a study of precipitation bias over the SGP and AMAZON sites and dis- cussions about the representativeness of SCM model results for the three-dimensional model. Is there a way to predict this representativeness before conducting SCM exper- iments? I would naively think that comparing the dynamical and physical tendencies from the three-dimensional model could be a way to do this.

This is an excellent question and this is something the authors feel should be explored in future work as it would serve as a large benefit to the community. While we do not have a conclusive answer to this question, at the end of section 5.2 we state:

Having comprehensive a priori knowledge on what particular biases and regimes could faithful be replicated within a SCM framework would be invaluable for GCM development and improvement. However, this is currently poorly understood and should be the subject of future work. 4. I think recent work by Smalley et al. (2019) on the SCM development and its use for parameterization testing and development deserves to be mentioned.

We agree and have added the following text to the introduction:

Smalley et al. (2019) use the SCM to construct a novel modeling framework that is forced by reanalysis to simulate a variety of environmental conditions in the subtropics to evaluate their parameterization suite. The authors thank reviewer 2 for taking the time to provide us with a careful review of our paper and for the positive feedback and helpful suggestions. Please see our responses below, that are reflected in our revised document and track changes.

**1. Section 2.3**

Line 103, for the current dynamical core for CESM, cite a paper or code documentation. Lines 105-7, the SCM provides the large-scale vertical transport or the full dynamical core does? Are you saying the Eulerian and SE dynamical cores calculate vertical transport differently? Maybe it's the phrase "dynamical core in the SCM" that is confus- ing because that doesn't make sense. Please make clear what the full 3D dynamical core calculates and what the SCM calculates and why that provides inconsistencies. Some explanation of how the codes are connected would be useful before delving into the details.

For the CESM dynamical core reference, we refer the reader to Lin and Rood (1997) and to the CAM5 technical document of Neale et al. (2012) for further information how the dynamical core was implemented into CAM5.

We realize that the explanation of the dynamical core and how the large scale forcing was computed was confusing. We have reworded this with the following statements:

"This is a problem because while the horizontal advection fields are provided by the IOP forcing files, the dynamical core is still responsible for computing the large scale vertical transport if this is not prescribed in the forcing file. In the E3SM/CAM SCM the only part of the dynamical core that is exercised is the computation of the large-scale vertical transport. The Eulerian and SE dynamical cores use two different methods for this computation, the former uses a simple Eulerian calculation while the later uses a semi-Lagrangian method. Therefore, the inherited SCM was inconsistent with the full GCM in regards to how the large scale vertical advection was computed."

Line 109, forward in time is not specific enough. I think you should specify the scheme with an explanation as to why the differences in the dynamical core time integration matters for the SCM. For example, I can imagine that a staged explicit scheme will not mesh well with the leapfrog scheme because then the time step stages are at different times. In the same vein, why does the SCM have to use what the 3D dynamical core uses? Do they only share info at the outer final time step from a staged method?

We changed "forward in time" to "the SE dynamical core uses a third-order five-stage explict Runge-Kutta (RK) method as described in Dennis et al. (2012)". We note that the SCM technically does not have to use the same dynamical core as the full model, but doing so helps to minimize the differences between the full model and the SCM. In this case, because the Eulerian core uses a leapfrog method, the physics time step has to be 2x the dynamics timestep. This is different than the full model in which the dynamics time step is equal to the physics timestep. Therefore, different timestep settings between the SCM and full 3D model may not provide a fully accurate representation of the full model in the SCM if certain schemes happen to be sensitive to timestep, as we note in section 2.3. Two paragraphs starting on line 112, the explanation of the SE grid needs more clarity. You need to explain how quadrilateral elements make up a sphere - a cube of faces that are then mapped to a sphere. You state that the SE grid must be initialized with a minimum resolution of points, I assume this is 1 per cube face? Do not use the HOMME/CAM-SE developers nomenclature of "ne4" unless you explain or cite. Then you can explain how you instantiate a low resolution version of SE and the SCM is only computed for one column (from one GLL point or for a point from the whole element?).

**We have revised this paragraph to be more clear. It now reads as:**

"Ideally, we want the SCM to be as close a proxy to the full GCM run as possible. Thus we upgraded the SCM dycore to use the same spectral-element (SE) dynamical core used by E3SM. Even though horizontal advection is prescribed in the SCM, the dycore still plays an important role for vertical advection. As described in Dennis et al. (2012), the SE dycore operates on quadrilateral elements whose Gauss-Lobatto-Legendre (GLL) quadrature points form the physics columns targeted by the SCM. Because there are many physics columns within each spectral element, it is impossible to initialize a single physics column when running a SE-dycore SCM. In this context the simplest way to initialize the SCM dycore is to ``trick" the model by initializing the dynamical core using a low resolution global configuration, but then only actually use a single physics column for our calculations. We do this using the lowest-resolution configuration supported by E3SM, which contains 96 elements and corresponds to a grid spacing of approximately 7.5 degrees at the equator. Our strategy requires slightly more memory (to initialize the whole dynamics grid) but no more computational expense than if we initialized just one column (because we only perform physics and vertical advection calculations on a single column)."

Rest of section 2.3, the advantages are just tacked on here. This should perhaps go somewhere else.

We agree these lines were out of place, and we moved these lines to the Summary and Discussion section.

2. Lines 150-154, this sentence is repetitive and not clear. The SCM used the Eulerian dynamical core before you upgraded it, ok. So you are just saying that the SCM re- quired files in the same Eulerian format or another strategy that is unclear. What does "it" at the end of the sentence refer to?

We removed the repetitive statement and the ambiguous reference to "it". The statement was clarified to mean that when running the E3SM SCM Replay option does not require post processing of the forcing files as does CAM's version of this functionality. This is mostly because forcing terms are slightly different and reside on different grid structure between dynamical cores.

3. End of section 4: I was hoping for some idea of what was done to make SCM work within CAM-SE. Just code bug fixes? A bunch of little things? Were any scientific changes made?

The main "crux" to get this to work was training the SE dycore to run with one column, which has been described in the second paragraph of section 4. Beyond that, it was a simple matter of

*isolating the dynamical core so that the only calculation it returns was for the large-scale vertical advection (i.e. no horizontal advection). At the end of section four we add:*

This was the main challenge towards being able to use the SE dynamical core in the SCM setting, after which we trained the SE dynamical to only calculate the large-scale vertical advection (i.e. no horizontal advection) in the column of interest if in SCM mode.

4. Several paragraphs starting on Line 233, I am on board for annual bias not showing up for SCM, but the daily precipitation bias for SCM is worse and different than GCM and obs. The meaning of this is implied in the next paragraph, line 249, that SCM can replicate early onset precipitation... but in a different thread of discussion, so that connection should be made more clear. Line 244, I don't see that, so you will need to plot bias as well. The statement "largest discrepancies occur during most pronounced bias" is not useful. 5. Line 268, what do you mean by "cannot afford to offer?" Do you mean it would take a long time to develop or it would make the SCM too expensive? Or both?

We realize the confusion here likely relates to poor wording choices in our original document. Thus, starting on line 233 we have clarified things with different wording choices and we also brought the discussion of the early onset of precipitation earlier when we discuss the bias results. Also, our choice of "cannot afford to offer" was a very poor choice of words. What we meant to say was that the "SCM is unable to provide insights" on this matter. We have modified the text to reflect this.

Minor comments: Will SCM change within version 2 of the E3SM? The use of "version 1" in the title implies that it is version dependent.

No. GMD requires us to note which version of the model we refer to in the title. However, in the conclusions we now state that we do not expect the E3SMv1 infrastructure to change with E3SMv2.

One minor exception to the good writing is the use of "which" and "that" (e.g. line 28,98,126, 225, 228, there may be more?) throughout and the use of tense (e.g. line 49, the paper will be organized, lines 111- 112 was then is).

Thank you. We feel we have addressed these issues.

Lines 42-44 and 56-58 are virtually ID, remove/adjust one of them. It's awkward where placed in the second instance.

This has been resolved by modifying the later statement.

Line 95 "The idealization switches added to the E3SM SCM framework includes" should use "include"

Resolved.

Line 98: Just say that some cases have idealizations turned on by default and move mention of specific cases later in the paper when they have been defined and ex- plained.

This has been resolved as suggested by the reviewer.

Line 147 "high degree" of accuracy is imprecise. Relative to what?

This has been reworded to read "to replicate a specific GCM column with only round-off error differences"

Line 174 first mention of nudging- should explain/define, and line 176 what are the impacts of nudging?

We have now defined nudging at first instance. We moved text from section 5.4 that discusses the impacts of nudging to this line, which feels more in place.

Line 179 Where was this already stated?

We removed this statement to avoid the inconsistency.

Line 204, did Xie19 implement or just describe this revised function? It is not clear from this description.

They implemented and analyzed the effects of the revised function. We changed the wording to make this more clear.

Line 255, once you have a fix you can see why the fix works. But what if you see a bias? How does it help you narrow down the source?

It is unclear to us what the reviewer is referring to here. This line is describing the MPACE case meteorology and setup and has not mentioned anything about a bias. Unless further clarification can be provided, nothing has been changed here.

Line 298, Though -> Although

Fixed.

Line 318, what do you mean by observational guidance?

Changed this to simply state "For observations we use..."

Line 332, "An example of this" should be "as an example of this"

Fixed

Figure 4 caption: second sentence, you mean the 2nd through 4th rows?

Yes, we have fixed this.

**The E3SM version 1 Single Column Model**

Peter A. Bogenschutz1, Shuaiqi Tang1, Peter Caldwell1, Shaocheng Xie1, Wuyin Lin2, and Yao-sheng Chen3,4 1Lawrence Livermore National Laboratory, Livermore, CA 2Brookhaven National Laboratory, Upton, NY

[revised manuscript text omitted]